# Arterial Blood Pressure Variability and Other Vascular Factors Contribution to the Cognitive Decline in Parkinson’s Disease

**DOI:** 10.3390/molecules26061523

**Published:** 2021-03-10

**Authors:** Anna Pierzchlińska, Magdalena Kwaśniak-Butowska, Jarosław Sławek, Marek Droździk, Monika Białecka

**Affiliations:** 1Department of Pharmacokinetics and Therapeutic Drug Monitoring, Pomeranian Medical University, Aleja Powstańców Wlkp 72, 70-111 Szczecin, Poland; monika-bialecka@post.pl; 2Division of Neurological and Psychiatric Nursing, Medical University of Gdansk, Aleja Jana Pawła II 50, 80-462 Gdansk, Poland; magdalena.butowska@gumed.edu.pl (M.K.-B.); jaroslaw.slawek@gumed.edu.pl (J.S.); 3Department of Neurology, St Adalbert Hospital, Aleja Jana Pawła II 50, 80-462 Gdansk, Poland; 4Department of Experimental and Clinical Pharmacology, Pomeranian Medical University, Aleja Powstańców Wlkp 72, 70-111 Szczecin, Poland

**Keywords:** white matter hyperintensities, dysautonomia, genetic polymorphisms, dementia, levodopa, renin-angiotensin system, orthostatic hypotension

## Abstract

Dementia is one of the most disabling non-motor symptoms in Parkinson’s disease (PD). Unlike in Alzheimer’s disease, the vascular pathology in PD is less documented. Due to the uncertain role of commonly investigated metabolic or vascular factors, e.g., hypertension or diabetes, other factors corresponding to PD dementia have been proposed. Associated dysautonomia and dopaminergic treatment seem to have an impact on diurnal blood pressure (BP) variability, which may presumably contribute to white matter hyperintensities (WMH) development and cognitive decline. We aim to review possible vascular and metabolic factors: Renin-angiotensin-aldosterone system, vascular endothelial growth factor (VEGF), hyperhomocysteinemia (HHcy), as well as the dopaminergic treatment, in the etiopathogenesis of PD dementia. Additionally, we focus on the role of polymorphisms within the genes for catechol-*O*-methyltransferase (*COMT*), apolipoprotein E (*APOE*), vascular endothelial growth factor (*VEGF*), and for renin-angiotensin-aldosterone system components, and their contribution to cognitive decline in PD. Determining vascular risk factors and their contribution to the cognitive impairment in PD may result in screening, as well as preventive measures.

## 1. Introduction

Parkinson’s disease (PD) is a neurodegenerative disorder with a wide spectrum of motor (bradykinesia, tremor, rigidity, loss of postural stability) and non-motor (cognitive decline, depression, dysautonomia, psychosis) symptoms. Parkinson’s disease dementia mostly affects the executive and visuospatial functions, as well as attention. However, the impairment in memory and language functions are less pronounced than in Alzheimer’s disease (AD) [1]. Dementia in PD is six times more common than in age-matched general population [2]. As a result of an 8-year prospective study, the cumulative prevalence of dementia was assessed to be 78% [3]. According to a 5-year prospective study in more than 400 patients with PD, the risk factors for dementia included older age, longer disease duration, later age-at-onset and higher daily levodopa (l-dopa) dosage [4]. In contrast to Alzheimer’s disease, vascular contribution to PD dementia is not so obvious. The available data demonstrate the multifactorial origin of PD dementia: The combination of pathological findings of Alzheimer-like and cortical Lewy-bodies, but also a possible role of vascular burden due to hyperhomocysteinemia and dysautonomia with abnormal blood pressure (BP) variability [5,6]. In this review paper, the authors present the current state of information about the role of BP variability and other vascular risk factors, as well as selected genetic polymorphisms, in the pathogenesis of PD dementia.

## 2. Blood Pressure Variability

In healthy individuals a compensatory mechanism, involving the sympathetic and parasympathetic systems, provides an adequate response of BP and cerebral perfusion to environmental (e.g., seasons, altitude), physical (e.g., posture), and emotional factors. The size and patterns of these BP variations constitute BP variability, which occurs within seconds and minutes (very short-term), along 24 h (short-term), between days (mid-term), months or even years (long-term; visit-to-visit BP variability) [7].

Numerous studies have shown a dysfunctional response of autonomic nervous system controlling blood pressure in PD [8,9]. Dysautonomia in PD is manifested, among others, in orthostatic hypotension (OH), supine hypertension or loss of circadian rhythm of BP, which may be the results of both autonomic nervous system dysfunction and pharmacological management, including dopaminergic medications.

Orthostatic hypotension is defined as a systolic pressure fall of at least 20 mmHg and/or diastolic pressure fall of 10 mmHg upon standing or passive tilting. Supine hypertension in PD is often associated with OH and causes an increase in BP when lying down—to at least 150 mmHg in systolic and/or 90 mmHg in diastolic BP. This phenomenon often remains unrecognized, thus an ambulatory 24-h BP monitoring provides a valuable insight into the BP alterations and enables non-pharmacological and pharmacological management. Supine hypertension can be additionally worsened by the drugs prescribed against OH. On the other hand, the treatment of hypertension may worsen OH, thus it should include non-pharmacological measures as a priority [8]. Four clinical trials have addressed OH in conditions with dysautonomia, including PD, and have posted results by the time of this review (clinicaltrials.gov: NCT00738062, NCT00782340, NCT01176240, NCT00633880). In all of them droxidopa—a precursor of norepinephrine—was compared with placebo. The results showed some discrepancies, as only two of the trials established significant clinical changes favoring droxidopa in terms of increasing upright systolic BP, ameliorating symptoms associated with OH and improving daily life activities (i.e., standing, walking) (NCT00782340, NCT00633880) [10].

Evaluating the circadian blood pressure profile in healthy subjects demonstrated a physiological decrease of arterial BP at night, while sleeping (dipping) [11]. This phenomenon is influenced by several factors, and we briefly describe only some of them. First of all, photoreceptor cells in retina detect the environmental darkness, subsequently, the information is conveyed via the suprachiasmatic nucleus—the central oscillator—to the pineal gland, which reacts by producing melatonin. Melatonin decreases BP by its impact on vasodilation, nitric oxide, and norepinephrine levels. Accordingly, the pivotal role in the circadian BP pattern is exerted by the sympathetic nervous system with norepinephrine and epinephrine having the lowest levels in the evening and the highest in the morning. Sleep itself affects BP, decreasing its values especially during the deep sleep stages [12]. Similarly, physical activity may change the circadian rhythm—people who work at night, are likely to have blunted nocturnal drop in systolic BP. Dipping depends also on the BP liability upon dietary sodium intake and endothelial function [13]. The loss of night BP reduction (non-dipping) has been recognized as a risk factor of cardiovascular pathologies, e.g., left ventricle hypertrophy [14]. Nocturnal hypertension is usually diagnosed when BP is higher than 120/70 mmHg [8]; whereas the term “non-dipping” relates to a relative difference between daytime and nighttime BP (i.e., <10% of difference). It is often associated with other cardiovascular autonomic abnormalities, such as OH and supine hypertension [15]. The 24-h BP monitoring showed that over 70% of PD patients did not have a proper circadian BP profile compared to 48% in the control group [16]. The analysis performed by Berganzo et al. revealed that the most frequent phenotype among the PD patients was non-dipper, while the number of the risers (mean arterial BP higher in the night than during the day) was the same as subjects having the physiological diurnal pattern [16]. Orthostatic hypotension was more prevalent in non-dippers and risers; those phenotypes administrated a higher dose of dopaminergic treatment; however, none of the associations reached statistical significance.

Neurocirculatory abnormalities in PD have been linked to cognitive impairment. In patients with early stage of the disease, who were not receiving dopaminergic medication, dementia was significantly more often in the subjects with OH or supine hypertension [17]. Strikingly, none of the patients suffering from both OH and supine hypertension was cognitively intact. The disruption of the nocturnal BP fall was related to worse visuospatial memory registration; however, mild cognitive impairment (MCI) or dementia were similarly frequent in dipper and non-dipper groups. Anang et al. carried out a prospective research with a comprehensive assessment of 80 PD patients who were cognitively intact at baseline [18]. After 4.4 years’ follow-up, higher BP at the entry and OH were predisposing factors for cognitive decline. Some of the cross-sectional studies reviewed by Udow et al. did not provide a correlation linking OH in PD with cognitive impairment assessed with Mini–Mental State Examination (MMSE) [19]. On the other hand, in analyses using more detailed neuropsychological tests OH was associated with worse performance. In one of the recent studies the riser pattern in PD was significantly correlated with dementia, whereas the dipper status was much less common in demented (3.7%) than non-demented (23.6%) subjects [20]. The authors found no association between dementia and non-dipper or extreme dipper (nocturnal BP lower >20% compared to daytime) phenotypes. Both OH and supine hypertension were significantly more prevalent in the dementia group. Thus, cardiovascular autonomic dysfunction can be considered as a plausible predictor of PD dementia.

## 3. The Role of Levodopa in Neurocirculatory Abnormalities and Cognitive Decline in Parkinson’s Disease

The dopaminergic medications used in the treatment of PD produce positive effects on PD motor manifestations, but at the same time may promote the development of side effects, e.g., dyskinesia may enhance autonomic disturbances [21]. Positron emission tomography (PET) and functional magnetic resonance imaging (fMRI) in experimental models and in PD patients have shown changes in dopaminergic synapses during long-term administration of l-dopa. The chronic use of l-dopa resulted in endothelial cell proliferation, promoted angiogenesis in the striatum and other structures of the basal ganglia in animals with l-dopa–induced dyskinesia [22]. Animal models have also evidenced that chronic l-dopa treatment induced expression of vascular endothelial growth factor (VEGF) in the basal ganglia in a dose-dependent manner [23]. VEGF exerts angiogenic activity and is essential for vasculogenesis. It can also enhance blood–brain barrier permeability, which in turn increases the bioavailability of l-dopa to the central nervous system (CNS). In the aforementioned model, VEGF was expressed predominantly in astrocytes and astrocytic processes around blood vessels. Administration of a VEGF signaling inhibitor blocked the appearance of markers of blood–brain barrier permeability (albumin extravasation) as well as of angiogenic response—nestin (a marker of immature endothelial cells) and VEGF expression. Furthermore, it was found that VEGF inhibition significantly attenuated the expression of dyskinesia in animal models with l-dopa-induced dyskinesia. Human tissue from the autopsies of chronically l-dopa-treated PD patients established increased nestin staining and *VEGF* mRNA expression in the striatum [23,24]. Pulsatile stimulation of dopamine receptors, which may be a result of abnormal l-dopa pharmacokinetics in the absence of efficient, endogenous regulatory mechanisms, is regarded as the main mechanism of peak-of-dose choreic dyskinesia (the result of D-receptor hypersensitivity or excessive dopamine release). Chronic l-dopa administration, apart from inducing angiogenesis and changes in blood–brain barrier permeability, stimulates vasodilation through D1 receptor agonist action, thus increasing regional blood flow [24].

According to a 5-year prospective study higher l-dopa dose was established as one of dementia risk factors in PD—the mean l-dopa dosage equivalent (LDE) was 514 mg per day, whereas non-demented patients received 316 mg of LDE per day [4]. Moreover, l-dopa-induced dyskinesia were correlated with the progression of cognitive decline, especially executive functions [25]. Daily changes in BP, dipper or non-dipper phenotype, OH and supine hypertension may modulate CNS blood flow, and thus affect l-dopa distribution, which may result in cognitive performance deterioration, and other side effects. Up to now, no studies have addressed the role of BP variability, especially nocturnal BP reduction, in the development of l-dopa complications, e.g., peak-of-dose dyskinesia due to impaired drug distribution.

Dopaminergic treatment was suggested to exacerbate dysautonomia in PD; l-dopa may worsen or cause OH [26], however, Goldstein et al. indicated that OH occurs independently from l-dopa treatment [27]. Although l-dopa is well known for its BP decreasing function [9,28], the available literature does not give a clear answer to the question whether it decreases supine hypertension [28,29]. There are some discrepancies on the role of dopamine agonists as well. Non-ergot derivatives of dopamine agonists possibly influence cardiovascular functions less, compared to ergot-derivatives [30]. Nevertheless, the side effects of non-ergot derivatives depend on the affinity to dopaminergic and α-adrenergic receptors [31]. Although non-ergot dopamine agonists were generally correlated with OH occurrence, rotigotine was showed to improve the abnormal 24-h BP pattern in PD patients [32]. On the other hand, rotigotine may increase the risk of the atrioventricular block. Pramipexole in turn, due to its high affinity towards α2-adrenergic receptors, may decrease the adrenergic tone and myocardial contractility, facilitating heart failure occurrence. Pramipexole may unmask a subclinical heart failure or exacerbate preexisting cardiovascular comorbidities [31]. Another non-ergot dopamine agonist—apomorphine—was suggested to prolong QT interval; however, no causality has been established [31,33]. Undoubtedly, the aforementioned cardiovascular side effects influence BP and may lead to the impairments within the cerebrovascular functions, such as episodes of hypo- and hyperperfusion.

The relationship between dopaminergic treatment and the BP variability needs to be further assessed. It seems important because disturbances in the diurnal BP pattern, especially nocturnal hypertension, are associated with an increased risk of cerebrovascular complications. Nevertheless, despite many side effects, levodopa is still the most often prescribed antiparkinsonian medication due to its effectiveness against motor symptoms. Non-ergot derivatives are in turn beneficial in treating non-motor symptoms, e.g., depression, sleep disorders, and nocturnal akinesia [31]. In most cases positive effects of dopaminergic medication (l-dopa and dopamine agonists) surpass potential side effects in the cardiovascular system, however, the risk must be always individually estimated.

## 4. White Matter Hyperintensities

Shimada et al. documented that an absent or insufficient nocturnal BP fall in elderly hypertensives was associated with a silent cerebrovascular damage seen in MRI as white matter hyperintensities (WMH) [34]. In recent years, WMH appearance has become the radiological marker of CNS ischemia. White matter hyperintensities are more frequently observed in subjects over 60 years of age with cardiovascular disease, diabetes, hyperhomocysteinemia, and other vascular risk factors [35,36]. Chronic ischemia being a consequence of BP variability was established as one of the possible factors leading to the development of WMH. In addition to a non-dipper phenotype, an excessive nocturnal BP fall of more than 20% compared to daytime BP values was found to be an important risk factor of WMH. The extreme dipper phenotype also was proven to be a risk factor for clinically silent brain ischemia episodes, particularly those associated with more extensive white matter damage, and can be related to nocturnal cerebral hypoperfusion [37,38]. Studies analyzing neurocirculatory abnormalities in PD demonstrated a correlation between WMH and OH [17,39]. Cerebral hypoxia along with hypoperfusion were proposed as mechanisms linking OH and WMH [40]. Indeed, hippocampal atrophy was correlated with WMH in the study performed by Sławek et al. [41]. Furthermore, WMH in PD was also associated with other cardiovascular dysautonomic features, as supine hypertension [17] or nocturnal hypertension [42]. Up to date, the only study examining nighttime BP fall and WMH in PD showed no correlation between non-dipping status and WMH [42].

According to the meta-analysis performed by Debette and Markus, WMH predicted faster decline in global cognitive performance and executive function [43]. In PD associated dysautonomia and dopaminergic medication-induced hypotension may be responsible for WMH formation and cognitive decline. In a prospective study in PD patients a significant relationship between increased WMH volume and cognitive burden was demonstrated. Moreover, the risk for the development of MCI in the course of PD was higher in the subjects with greater WMH [44]. Subsequent studies supported the finding, demonstrating a significant correlation between WMH and MCI or dementia in PD [45,46]

Cardiovascular dysautonomia in PD, along with dopaminergic therapy, leading to episodes of cerebral hypo- and hyperperfusion, may be one of the most important pathophysiological mechanisms leading to WMH and, presumably, dementia.

The studies on the impact of BP variability on cognition and WMH in PD are summarized in Table 1.

## 5. Metabolic and Vascular Risk Factors

The etiology of cognitive impairment in PD presumably involves other than dopamine-dependent mechanisms, from which factors affecting the cerebrovascular status seem highly probable (Figure 1) [61]. Hypertension, diabetes mellitus and hypercholesterolemia lead to structural changes in the vessels [52], hypoperfusion, endothelial disorders, and altered blood–brain barrier permeability [70]. This results in cerebrovascular pathologies, seen as WMH—a marker of cognitive deterioration [52].

Vascular risk factors and their significance in PD and PD dementia have been investigated in several analyses, one of which was a cohort study on Finnish population assessing the impact of diabetes on PD occurrence. The hazard ratio of PD among patients with type 2 diabetes at baseline was estimated as 1.83 (95% CI 1.21–2.76) [71]. However, in the Nurses’ Health Study and the Health Professionals Follow-up Study the relationship between PD and self-reported diabetes was not confirmed [72]. In the same analysis self-reported history of hypertension or high cholesterol levels were not established as PD risk factors either. In the meta-analysis performed by Cereda et al. diabetes was correlated with PD, although the diagnosis of idiopathic PD without imaging techniques in some studies seemed to be uncertain, since diabetes might lead to microvascular complications with parkinsonism as clinical outcome [73]. The impact of diabetes on PD dementia in a meta-analysis by Xu et al. was not confirmed, in contrast to hypertension (OR 1.57, 95% CI 1.11–2.22) [74]. The relationship between hypertension and cognitive performance was additionally found in Doiron et al. research [70]. In this prospective study, the history of hypertension was followed by lower Z-scores on immediate and delayed free recall, recognition, and verbal fluency tests. However, the numbers of years with hyperlipidemia did not correlate with a change in any of the Z-scores at the 24-month follow-up. In a prospective research, hypertension was correlated with an increased risk of the MCI development in PD patients, whereas MCI and WMH predicted the conversion to PDD [52]. No impact of hypertension on the progression from MCI to PDD was established. Thus, it is likely that hypertension contributes to increasing WMH, which in turn modulates the risk of dementia. On the other hand, vascular factors (e.g., smoking status, body mass index, hypertension, and diabetes)—calculated together into Framingham General Cardiovascular Disease Risk Score—increased the risk of both MCI and dementia in PD [61]. Homocysteine (Hcy) is an established risk factor for vascular damage. Additionally, it has been correlated with neurodegeneration related to oxidative stress, calcium accumulation and apoptosis [75]. An elevated concentration of homocysteine (hyperhomocysteinemia: HHcy; value over 15 µmol/L of blood) was more frequently observed in PD dementia than in PD patients without dementia [41]. Hcy elevated levels might result from l-dopa methylation by catechol-*O*-methyltransferase (COMT), but the authors assessed the correlation between Hcy levels and l-dopa dose as weak (coefficient of determination R2 = 0.12). However, an increase in Hcy level was established for the treatment with duodenal levodopa gel (duodopa), due to its formulation which negatively affects the absorption of B6 vitamin and folate, necessary for reducing Hcy levels [76,77]. The correlation between Hcy and cognitive status in PD was demonstrated in a meta-analysis performed by Xie et al., with suggestion that the detrimental effect of HHcy could be mitigated with folate and/or B12 vitamin supplementation [78].

The plasma and tissue renin–angiotensin–aldosterone (RAA) system has recently emerged as another risk factor for PD and PD dementia. It plays a key role in controlling BP as well as water and electrolytes homeostasis. The conversion of angiotensinogen to angiotensin II (AngII) is catalyzed by renin, and then angiotensin converting enzyme (ACE). The resultant AngII binds to type 1 or 2 angiotensin receptors (AT1R, AT2R), expressed among others in peripheral vessels, heart, kidneys and CNS. Initially, the effects of the RAA system in CNS were considered to be mediated only through circumventricular organs, since active products of this system do not cross the blood–brain barrier [79]. However, at this moment it is known that the brain has an independent local RAA system, and the existence of the intracellular RAA system has been confirmed. As demonstrated in the experimental model, angiotensinogen in CNS is produced by astrocytes with a limited contribution from neurons, and central AngII levels may be even higher than the peripheral concentration [80]. The role of the RAA system in the nigro-striatal pathway seems to be of particular interest. Dopamine deficiency leads to compensatory RAA system hyperreactivity and an increased expression of AT1R and AT2R [81]. In neurons, interaction of AngII with its receptor AT1 leads to activation of neuronal NADPH oxidase and production of reactive oxygen species (ROS). This triggers oxidative stress and inflammatory processes that are additionally enhanced by ROS from other sources i.e., mitochondria or activated microglia. Free radicals from the latter are released extracellularly, and consequently, may lead to progressive damage to dopaminergic neurons [79].

Experimental and clinical evidence demonstrates beneficial effects of AT1 receptor blocking or converting enzyme inhibition on RAA system function, and dopamine concentration in striatum, as well as a reduction in inflammatory processes [79,82]. The RAA system is closely linked to VEGF synthesis, and like AngII, VEGF stimulates angiogenesis, affects permeability of blood vessels and blood brain barrier function. Janelidze et al. established the correlation between cerebrospinal fluid (CSF) biomarkers of angiogenesis and PD with and without dementia [40]. Levels of VEGF, placental growth factor (PlGF), and one of VEGF receptors (VEGFR-2) were significantly higher in both demented and non-demented PD patients comparing to the control group. However, no difference was observed between PD and PD dementia groups. In another study, the CSF level of VEGF was considerably higher in PD dementia (grouped together with patients with Lewy bodies dementia) than in the control subjects [83]. VEGF was suggested as a potent trigger of vascular leakage and blood–brain barrier dysfunction, which is in line with elevated endothelial cell nuclei and vessels in the substantia nigra pars compacta of PD patients found in post-mortem studies [23,84]. Moreover, it was noted that patients with diastolic OH had increased levels of VEGF and PlGF in comparison with subjects without OH. Insufficient autoregulation of cerebral blood flow in PD patients results in hypoxia-induced VEGF signaling and, consequently, in an angiogenic response [40].

Some proteins from the VEGF family demonstrate neuroprotective activity (VEGF-B), as revealed in diencephalon cell cultures exposed to rotenone (widely used as an experimental model of PD). The neuroprotective effect of VEGF on 6-hydroxydopamine (6-OHDA) treated dopaminergic neurons was reported to be mediated by both direct and indirect vascular and neuronal mechanisms. The neuroprotective effect of VEGF in PD models was dose-dependent, i.e., the low dose protected dopaminergic neurons, whereas the high dose induced severe brain edema [85]. Excessive activation of the tissue and/or plasma RAA system resulting from dopamine deficiency may lead to the fostering of processes damaging dopamine neurons. This in turn may increase the risk of l-dopa-induced side effects (e.g., dyskinesia) due to both increased neurodegeneration and changes in l-dopa distribution within CNS (pulsatile dopamine receptor stimulation).

Muñoz et al. demonstrated that administration of candesartan, AT1 receptor blocker, significantly reduced dyskinesia in the 6-OHDA induced, experimental model of PD [86]. In animals with dyskinesia, higher concentrations of VEGF and IL-1β within striatum and substantia nigra were found. Furthermore, animals treated with candesartan along with l-dopa were characterized by lower levels of VEGF, IL-1β, and less severe dyskinesia compared to animals receiving only l-dopa. l-dopa monotherapy reduces the activity of the RAA system. It is likely that the effects of l-dopa on angiogenesis in the course of chronic treatment cannot be eliminated, but can be mitigated by AT1R blockade. Moreover, the reduction of l-dopa-induced dyskinesia did not diminish l-dopa effectiveness against motor symptoms. Taking into consideration the aforesaid facts that long-term l-dopa treatment is a risk factor for developing PD dementia and that l-dopa-induced dyskinesia correlate with cognitive decline, it is possible that blocking AT1R may decrease the risk of cognitive decline.

A recent prospective study assessed the impact of several angiotensin II-stimulating (thiazides, dihydropyridine calcium channel blockers, AT1R blockers) and angiotensin II-inhibiting (ACE inhibitors, β-blockers, nondihydropyridine calcium channel blockers) antihypertensive drugs on the risk of dementia in older people (70–78 years at baseline) [87]. Subjects administrating AngII-stimulating antihypertensives had significantly lower risk of developing dementia compared to AngII-inhibiting drugs, independently of systolic BP or the baseline history of stroke, diabetes or CV diseases. As for the individual types of antihypertensives, the only significant decrease of dementia risk was associated with dihydropyridine calcium channel blockers. The beneficial role of upregulating AngII may result from the neuroprotection connected with AT2R. On the other hand, as ACE was showed to degrade amyloid-β (Aβ) in experimental models, the use of ACE inhibitors may promote Aβ plaque formation [87,88]. Regardless of the pivotal role of Aβ in AD, its deposition in PD was associated with cognitive decline [89]. However, in the aforementioned prospective study, the incidence rate of dementia in the group using ACE inhibitors did not differ from other types of antihypertensive drugs [87]. Moreover, Zhuang et al. showed in their meta-analysis that ACE inhibitors, as well as AT1R blockade, lowered the risk of AD, and AT1R blockade additionally decreased the risk of cognitive impairment associated with age [90]. Thus, the possible involvement of ACE in cognitive impairment and the effects exerted by ACE inhibitors seem to be far from being elucidated.

## 6. Genetic Factors

Although several causative genes for PD have been found, they scarcely account for some of the familial cases, whereas for the idiopathic form of PD and PD dementia, only susceptibility genes have been established, though not without inconsistency. Among the analyzed susceptibility genes are these coding for catechol-*O*-methyltransferase (*COMT*), apolipoprotein E (*APOE*), vascular endothelial growth factor (*VEGF*) and for renin–angiotensin–aldosterone system.

### 6.1. COMT

Catechol-O-methyltransferase (COMT) catalyzes the transfer of a methyl group from S-adenosyl-methionine to a hydroxyl group on a catecholamine (e.g., dopamine, norepinephrine, or catechol estrogen), thus regulating dopamine level in the prefrontal cortex, and is also a crucial enzyme involved in L-dopa metabolism [91,92]. Furthermore, the methylation of L-dopa is followed by the production of S-adenosylhomocysteine, hydrolyzed to homocysteine, which in turn may be responsible for HHcy in l-dopa medicated PD patients. However, homocysteine levels can be reduced by concurrent administration of COMT inhibitors in PD patients [93]. A common polymorphism in *COMT* gene—a G→A substitution (rs4680) results in an amino acid codon alteration (Val158Met in the membrane-bound form of COMT), and according to Chen et al., significantly lower enzymatic activity in homozygous *COMT*-Met in comparison with homozygous *COMT*-Val in post-mortem human prefrontal cortex tissues [91]. The dopaminergic imaging study demonstrated that Met homozygotes were characterized by an increased level of caudate striatal dopamine transporter (DAT), and a slightly protective effect on dementia was found for homozygous *COMT*-Met carriers [94]. Additionally, in a prospective analysis, Val homozygotes were at greater risk of developing PD-MCI than the other *COMT* genotypes, although the correlation with PD dementia was not observed [95]. Interestingly, Williams-Gray et al. [96] demonstrated in their longitudinal CamPaIGN study a varying impact of *COMT* alleles in relation to the disease duration. In “early” disease (<1.6 years), there was a significant decrease in the Tower of London (TOL; test of planning) scores with an increasing number of Met alleles, whereas no effect was observed in the “later” disease group (>1.6 years). Moreover, after 5.2-year follow-up, Met homozygotes were more likely to improve the TOL test score, in contrast to Val homozygotes or heterozygotes. The observed difference is possibly a consequence of an inverted U-shaped relationship between dopamine levels and prefrontal function, assessed mainly by working memory [96].

The *COMT* polymorphism seems to affect Hcy total level as well. Tunbridge et al. established that Val carriers had 1 µmol/L higher Hcy plasma values compared with Met homozygotes [92]. Additionally, the effect correlated with a polymorphism in methylenetetrahydrofolate reductase (*MTHFR* 677C > T; rs1801133) gene, which is commonly analysed with regard to Hcy/folate metabolism, resulting in the highest Hcy level among PD patients with both *MTHFR* 677TT and *COMT*-Val genotypes [92,97]. This indicates a necessity of COMT inhibitors administration, especially in patients at the greatest genetic risk for developing HHcy. However, in another study, significantly higher Hcy plasma levels were observed only in PD patients with *MTHFR* 677TT and low activity-determining *COMT* genotypes (based on the genotyping of four *COMT* single nucleotide polymorphisms—SNPs: rs4680, rs6269, rs4633 and rs4818) at the same time. The authors showed a correlation between Hcy levels and PD dementia, although none of the analyzed polymorphisms had an impact on cognitive impairment [97].

### 6.2. APOE

Apolipoprotein E (apoE) is associated with cerebrovascular and neurodegenerative diseases, such as late onset AD and PD [98,99]. The *APOE* gene polymorphism is identified in the form of three major alleles *APOE2*, *APOE3,* and *APOE4*, which determine three protein isoforms (E2, E3, and E4, resp.) and six possible genotypes (*e2/e2*, *e2/e3*, *e2/e4*, *e3/e3*, *e3/e4*, and *e4/e4*). Several studies showed that the *e2* allele is associated with a higher risk of PD [100,101], whereas in others, the *e4* allele was a risk factor. However, the data is inconsistent [102,103]. It was suggested that *APOE4* expression exerts detrimental effects on the cerebrovascular system, including blood–brain barrier impairments [104]. Indeed, Bell et al. showed that mice expressing human *APOE4* had altered blood–brain barrier permeability and reduced cerebral blood flow compared with animals expressing *APOE2* or *APOE3* [105]. In addition, apoE mediates the clearance of Aβ across blood–brain barrier, through binding to its liver receptor (low-density lipoprotein receptor-related protein-1—LRP1), and that *APOE4* allele contributes to cerebral accumulation of Aβ [105].

In a study performed by Janelidze et al., blood–brain barrier permeability characterized by the use of the cerebrospinal fluid/plasma albumin ratio (Qalb) differed significantly in groups with dementia (AD, dementia with Lewy bodies or PD dementia, vascular dementia or frontotemporal dementia) compared to healthy controls, although no impact of *APOE4* allele on Qalb was found [83]. Nevertheless, Qalb seemed to correlate with CSF biomarkers of angiogenesis or endothelial damage, i.e., intracellular adhesion molecule 1 (ICAM-1), vascular cell adhesion molecule 1 (VCAM-1) and VEGF, in all diagnostic groups. Those results support the role of the blood–brain barrier leakage in dementia, including PD dementia.

Despite some discrepancy in the studies assessing the role of *APOE4* in cognitive status of PD patients, possibly due to different diagnostic criteria for dementia [106], two meta-analyses [101,107] showed an over-representation of *APOE4* carriers in PD dementia groups compared to cognitively normal PD patients. Additionally, the most recent meta-analysis revealed that *APOE4* was a risk factor for PD dementia development regardless of the population origin [107].

### 6.3. VEGF

Variability in *VEGF* expression, induced by specific *VEGFA* variants, is involved in angiogenesis-related disorders. At least 30 SNPs in this gene have been described, and some SNPs can alter VEGF serum levels. Three common SNPs, namely −2578C/A in the promoter region (rs699947), −634C/G in the 5-untranslated region (rs2010963) and +936C/T in the 3-untranslated region (rs3025039) are related to VEGF protein production [108,109], although no association between VEGF serum level and PD has been established so far [110]. Some *VEGF* SNPs have been examined as susceptibility factors to AD. Del Bo et al. found a correlation between AD and −2578A/A and −1198C/T genotypes [111]. Although the VEGF serum level did not differ between AD patients and controls, increased values were correlated with *VEGF* polymorphisms, which had previously been described as associated with AD. Furthermore, a link between the severity of cognitive impairment and VEGF level was determined in Alvarez et al. study—the protein values were higher in AD than in MCI patients and in the controls [112]. The number of wild-type *VEGF* −2578C alleles was positively associated with total grey matter volume, total white matter volume and total arterial blood volume in young adults [113]. Considering that brain atrophy, thus smaller brain volume, correlates with cognitive decline in in PD [114], this can suggest a protective role of the *C* allele.

In the studies on *VEGF* genetic polymorphisms in PD patients, rs3025039 was the only *VEGF* polymorphism determined to correlate with PD development [110,111,115]. *VEGF* gene expression interacted with the genetic susceptibility factor for PD dementia. i.e., *APOE4*, on global cognition in AD, but not on AD neuropathology suggesting independence of the interaction from AD neuropathology [116]. However, no research has analyzed the impact of *VEGF* polymorphisms on cognitive performance or dementia in PD.

### 6.4. RAA System Genes

The neuroprotective effects of ACE inhibitors or AT1R antagonists observed in animal PD models suggest that abnormalities in the RAA system may promote the PD development [117]. Over the past several years, numerous polymorphic loci in genes encoding various components of the RAA system were defined, e.g., an insertion/deletion polymorphism in angiotensin converting enzyme gene (*ACE*) [118] or SNPs in angiotensin II receptor type 1 (*AGTR1*) [119], angiotensin II receptor type 2 (*AGTR2*) [120] or in angiotensinogen (*AGT*) [121] genes.

The insertion/deletion polymorphism in the angiotensin-converting enzyme gene (*ACE* I/D) was the first in the RAA system to be examined as a potential susceptibility factor in PD [122]. Though no association was found by Mellick et al. in an Australian population [122], or Pascale et al. in an Italian one [123], a study in a Chinese population showed an increased frequency of *DD* genotype in PD compared to the control group [124]. This discrepancy may result from differences in *ACE* I/D polymorphism frequency among the respective populations, since the prevalence of the *D* allele is estimated to be 50–58% in Caucasians and 35–39% in a Chinese population [124].

On the other hand, in a Swedish study the *II* homozygotes had a two-fold higher risk of dementia (including AD and vascular dementia) than the other *ACE* I/D carriers [125]. However, the polymorphism was not associated with dementia in the follow-up. Similarly, the *CC* genotype of *AGTR1* rs5186 was associated with dementia only at baseline [126]. Three other SNPs in *AGTR1* (rs2638363, rs1492103, rs2675511) were signs of worse episodic memory performance in a 4 years’ follow-up and additionally correlated with a hippocampal atrophy in older adults [127].

Purandure et al. reported a link between *ACE* I/D polymorphism and WMH in patients of Caucasian origin with AD or vascular dementia [128]. White matter hyperintensities were more severe in patients carrying the *DD* than *ID* (*p* = 0.01) or *II* genotype (*p* = 0.009) and the correlation remained significant after a correction for cardiovascular risk factors, thus suggesting other mechanisms contributing to WMH development. It is worth mentioning that carriers of both *APOE4* genotype and *ACE I* allele were at higher risk of developing late-onset AD [129].

Other *ACE* polymorphisms (rs4362), along with the SNP in the angiotensinogen gene (*AGT* rs699) and the SNP in angiotensin II receptor type 1 gene (*AGTR1* rs5182) were evaluated in older Australians [130]. A male-only relationship between *AGT* rs699 and *ACE* rs4362 polymorphisms and WMH was found, independently of hypertension. Moreover, the authors reported a synergistic effect of *AGT* rs699 and *AGTR1* rs5182 on WMH. Although it was established that *ACE* I/D polymorphism accounts for 47% of the variation in ACE serum level [131], other genetic polymorphisms, including synonymous ones, can affect gene expression and protein synthesis. Therefore, they may exert an impact on WMH and cognitive functions.

The available literature does not provide a comprehensive assessment of variability in genes encoding the RAA system components and their associations with BP variability in the course of PD or cognitive decline in PD (Table 2).

## 7. Conclusions

There is a growing demand to determine factors predisposing to the development of PD dementia. The impact of abnormal circadian BP variability observed in PD patients seems to contribute to WMH, which in turn may be a radiological marker for cognitive decline. Many of the presented factors, correlating with WMH hyperintensities and/or cognitive decline in PD, may and should be treated as far as possible (Figure 2). Hypertension, OH, supine hypertension and the absence of nocturnal BP fall can be diagnosed by an ambulatory 24-h BP monitoring and then managed by both nonpharmacological and pharmacological measures. Similarly, the impact of hyperhomocysteinemia—a metabolic risk factor for dementia in PD—may be possibly alleviated by more frequent blood concentration assessments and folate and/or B12 vitamin supplementation. As for the genetic risk factors, they may serve as markers of cognitive decline in PD or indicate a future direction for specific treatment, e.g., AGT1 receptor blockers and inhibitors of the RAA system. In summary, knowledge on vascular risk factors and their contribution to the cognitive impairment in PD may result in prophylaxis and better screening methods. However, this matter needs to be addressed in future studies, including clinical trials.

## Figures and Tables

**Figure 1 molecules-26-01523-f001:**
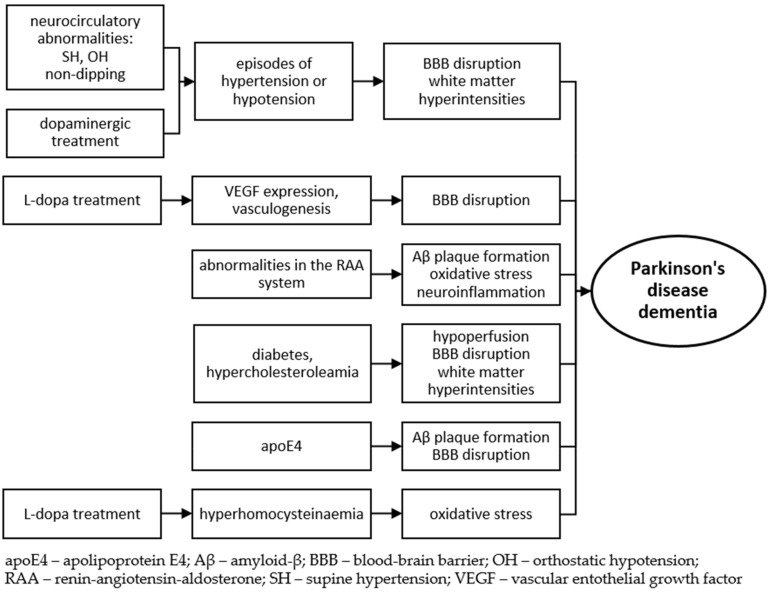
Possible vascular-related mechanisms leading to dementia in Parkinson’s disease.

**Figure 2 molecules-26-01523-f002:**
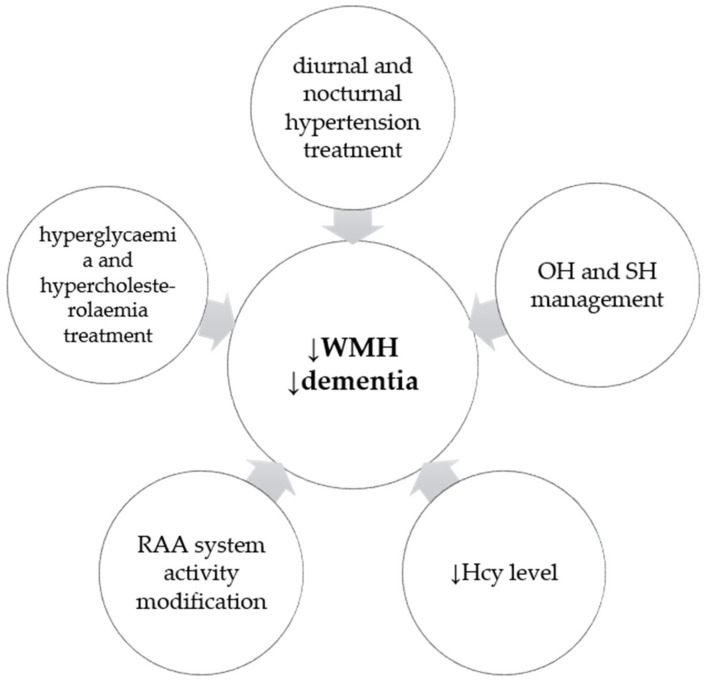
Possible measures decreasing the risk of developing WMH, and/or dementia, in Parkinson’s disease.

**Table 1 molecules-26-01523-t001:** Studies on the impact of blood pressure (BP) variability and white matter hyperintensities (WMH) on cognitive performance in Parkinson’s disease (PD).

Blood Pressure Variability			
Authors (year), Type of the Study	Subjects	Factor	Outcome
Oka et al. (2020) [47], retrospective study	PD = 75 (de novo) DLB = 24	Circadian blood pressure variability. Supine hypertension.	Better performance assessed with MMSE positively associated with the percentage of nocturnal BP fall in PD. No correlation with SH. Nocturnal BP fall (%) positively associated with better performance in Frontal Assessment Battery. SH negatively correlated with Frontal Assessment Battery.
Sforza et al. (2018) [48], retrospective study	PD = 28	Orthostatic hypotension.	In upright position, PD-OH(+) performed worse at the Stroop’s test word reading time and number of errors at the interference section compared to PD-OH(−).
Tanaka et al. (2018) [20], retrospective study	PD-NCI = 110 PDD = 27	Circadian blood pressure variability. Orthostatic hypotension.	The riser pattern associated with dementia. Coexistence of the riser pattern and OH more associated with dementia than the riser pattern alone.
Centi et al. (2017) [49], retrospective study	PD-OH(+) = 18 PD-OH(−) = 19 controls = 18	Orthostatic hypotension.	Upright posture correlated with the deficits in sustained attention and response inhibition, reduced semantic fluency and verbal memory in the PD-OH(+). OH correlated with the deficits in executive function, memory and visuospatial function.
Anang et al. (2014) [18], prospective study	PD = 80 (dementia free at baseline)	Orthostatic hypotension.	OH strongly associated with dementia risk.
Pilleri et al. (2013) [50], retrospective study	PD = 48	Orthostatic hypotension.	PD-OH(+) performed significantly worse in sustained attention, visuospatial and verbal memory, compared with PD-OH(−).
Kim et al. (2012) [17], retrospective study	PD-NCI = 25 PD-MCI = 48 PDD = 14	Circadian blood pressure variability, supine hypertension and orthostatic hypotension.	The OH group had more severe impairment in verbal immediate/delayed memory. Dementia significantly more prevalent in patients having OH, SH or OH + SH. Non-dipping not associated with cognitive impairment.
Allcock et al. (2006) [51], retrospective study	PD-OH(+) = 87 PD-OH(−) = 88	Orthostatic hypotension.	OH(+) subjects worse in sustained attention and visual episodic memory. OH not associated with the MMSE score, the prevalence of dementia, or the simple and choice reaction times, working memory or long term memory.
**White Matter Hyperintensities**			
**Authors (Year)**, **Type of the Study**	**Subjects**	**Factor**	**Outcome**
Nicoletti et al. (2021) [52], prospective study	PD-NCI = 84 PD-MCI = 55	WMH	WMH was a predictor of PDD development.
Huang et al. (2020) [53], retrospective study	Early PD: PD-NCI = 81 PD-MCI = 94	WMH	PD-MCI associated with the periventricular WMH but not with total WMH. Periventricular WMH associated with worse executive function and visuospatial function.
Ramirez et al. (2020) [54], retrospective study	PD = 139	WMH	WMH negatively associated with global cognition.
Dadar et al. (2020) [55], prospective study	PD = 50 controls = 45	WMH	No correlation between WMH and total Dementia Rating Scale. Greater WMH burden in patients diagnosed with dementia at 36 months.
Linortner et al. (2020) [56], retrospective study	PD = 85 controls = 18	WMH	Dementia and executive impairment significantly more prevalent in PD patients with WMH than without WMH. WMH associated with worse performance in Symbol Digit Modalities and Stroop tests.
Lee et al. (2020) [57], retrospective study	PD = 136 (de novo)	WMH	Performance in language function, frontal/executive and visual memory associated with the severity of WMH.
Chahine et al. (2019) [58], prospective study	PD = 141 controls = 63	WMH	Annual rate of change in global cognition correlated with WMH. Higher temporal WMH associated with greater decline over time in verbal memory.
Hanning et al. (2019) [59], prospective study	Drug-naïve: PD-NCI = 79 PD-MCI = 29 controls = 107	WMH (volume and CHIPS score)	No association between global or localised WMH and cognitive decline, both cross-sectional and longitudinal.
Pozorski et al. (2019) [60], prospective study	PD = 29 controls = 42	WMH	Greater regional and global WMH at baseline more strongly associated with lower executive function in PD than in controls. Increased regional WMH more strongly associated with impaired memory performance in PD relative to controls. Longitudinally, no associations between WMH and cognitive change.
Stojkovic et al. (2018) [61], retrospective study	PD-NCI = 49 PD-MCI = 61 PDD = 23	WMH	PDD patients had significantly greater whole brain WMH than PD-NCI subjects.
Dadar et al. (2018) [62], prospective study	PD = 365 (de novo) controls = 174	WMH	PD subjects with greater WMH had significantly more severe cognitive decline than PD subjects with low WMH load or controls with high WMH load.
Ham et al. (2016) [63], retrospective study	PD = 171 (non-demented)	WMH	Total WMH and deep WMH associated with worse performance in semantic fluency.
Mak et al. (2015) [64], retrospective study	PD-NCI = 65 PD-MCI = 25	WMH	Greater total and periventricular WMH in PD-MCI than in PD-NCI. Spatial distribution of WMH associated with global cognition, performance on the Frontal Assessment Battery and Fruit Fluency.
Sunwoo et al. (2014) [45], prospective study	PD-NCI = 46 PD-MCI = 65	WMH (volume and CHIPS score)	The progression from PD-MCI to PDD correlated with WMH volume and CHIPS score. In PD-MCI patients WMH volume and CHIPS score associated with longitudinal decline in general cognition, semantic fluency and Stroop test scores.
Kandiah et al. (2013) [44], prospective study	PD-NCI = 67 PD-MCI = 24	WMH	PD-MCI patients had significantly greater volume of periventricular and deep subcortical WMH than PD-NCI. Regional WMH significantly greater among PD-MCI in the frontal, parietal and occipital regions.
Sławek et al. (2013) [41], retrospective study	PD-NCI = 135 PDD = 57 controls = 184	WMH	WMH significantly greater in PDD than PD-NCI group.
Kim et al. (2012) [17], retrospective study	PD-NCI = 25 PD-MCI = 48 PDD = 14	WMH (CHIPS score)	The severity of WMH in the periventricular and subcortical white matter higher in PDD than in PD-NCI or PD-MCI. No difference in WMH between PD-NCI and PD-MCI.
Shin et al. (2012) [65], retrospective study	PD-NCI = 44 PD-MCI = 87 PDD = 40	WMH (CHIPS score)	The CHIPS score significantly higher in PDD than in PD-NCI or PD-MCI. WMH negatively associated with performance in MMSE. The CHIPS score correlated with the performance in contrasting programme and forward digit span tests.
Lee et al. (2010) [66], retrospective study	PD-NCI = 11 PD-MCI = 25 PDD = 35	WMH	Greater total and periventricular WMH in the PDD group compared to PD-MCI and PD-NCI groups. No difference in WMH between PD-MCI and PD-NCI groups.
Dalaker et al. (2009) [67], retrospective study	Drug-naïve: PD-NCI = 133 PD-MCI = 30 controls = 102	WMH	No differences between the groups in total volume or spatial distribution of WMH. No correlation between WMH and cognitive functions.
Beyer et al. (2006) [68], retrospective study	PD-NCI = 19 PDD = 16 controls = 20	WMH	PDD group had significantly more WMH in deep white matter and periventricular areas than the PD-NCI group.
**Interaction between WMH and BP Variability**			
**Authors (year)**, **type of the study**	**Subjects**	**Factor**	**Outcome**
Dadar et al. (2020) [69], prospective study	PD = 365 (de novo) controls = 174	WMH Orthostatic hypotension.	A correlation between WMH burden and worse Montreal Cognitive Assessment (MoCA) score in PD over time. WMH linked with diastolic OH. Direct effect of diastolic OH on the rate of cognitive decline via WMH burden.
Oh et al. (2013) [39], retrospective study	PD = 117	WMH Orthostatic hypotension, supine hypertension.	Orthostatic hypotension and supine hypertension correlated with WMH score.
Oh et al. (2013) [42], retrospective study	Drug-naïve: PD = 129	WMH Circadian blood pressure variability.	Nocturnal hypertension associated with WMH in the basal ganglia. No influence of the non-dipping pattern on WMH. Nighttime systolic BP closely correlated with WMH.
Kim et al. (2012) [17], retrospective study	PD-NCI = 25 PD-MCI = 48 PDD = 14	WMH (CHIPS score) Circadian blood pressure variability, supine hypertension and orthostatic hypotension.	WMH significantly more severe in patients having OH, SH or OH + SH. No difference in WMH between the dippers and non-dippers.

BP—blood pressure; DLB—dementia with Lewy Bodies; CHIPS—Cholinergic Pathways Hyperintensities Scale, measures the extent of WMH in the periventricular and subcortical white matter; MCI—mild cognitive impairment; MMSE—Mini-Mental State Examination; NCI—no cognitive impairment; OH(+)—with orthostatic hypotension; OH(−)—without orthostatic hypotension; PDD—PD dementia; WMH—white matter hyperintensities.

**Table 2 molecules-26-01523-t002:** Genetic factors influencing vascular functions and cognitive decline in PD.

Gene	Name of the Protein	Function of the Protein	Role in Cognitive Decline	References
*COMT*	catechol-*O*-methyltransferase	metabolism of catecholamines and l-dopa, involved in Hcy synthesis	polymorphism associated with Hcy overproduction	[92]
*APOE*	apolipoprotein E	component of several lipoproteins	*APOE4* variant linked to Aβ accumulation and BBB disruption	[105]
*VEGF*	vascular endothelial growth factor	angiogenic activity, essential in vasculogenesis.	microvascular pathologies, BBB disruption	[23]
*MTHFR*	methylenetetrahydrofolate reductase	Hcy and folate metabolism	polymorphism correlated with HHcy	[132]
*ACE*	angiotensin converting enzyme	catalyzes AII synthesis; its inhibitors lower BP	Aβ degradation; polymorphism associated with WMH	[87,130]
*AGT*	angiotensinogen	precursor of all components within the RAA system	polymorphism associated with WMH	[130]
*AGTR1*	angiotensin receptor type 1	AII receptor	oxidative stress and neuroinflammation; polymorphisms associated with hippocampal atrophy	[79,127]
*AGTR2*	angiotensin receptor type 2	AII receptor	possible neuroprotection	[87]
*HMGCR*	HMG-CoA reductase	rate-controlling enzyme in the cholesterol synthesis pathway	polymorphism decreases cholesterol production	[133]

AII—angiotensin II; Aβ—amyloid-β; BBB—blood–brain barrier; Hcy—homocysteine; HHcy—hyperhomocysteinemia; HMG-CoA—3-hydroxy-3-methylglutaryl-CoA; RAA—renin-angiotensin-aldosterone; WMH—white matter hyperintensities.

## Data Availability

Not applicable.

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
