# Peer review of "Arterial Blood Pressure Variability and Other Vascular Factors Contribution to the Cognitive Decline in Parkinson’s Disease"

_molecules, 2021, doi:10.3390/molecules26061523_

Round 1

Reviewer 1 Report

Thank you for submitting the manuscript “Arterial Blood Pressure Variability and Other Vascular Factors Contribution to the Cognitive Decline in Parkinson’s Disease” to Molecules.

The review article is coherent and has a large volume of references, demonstrating that the authors have knowledge of the topic. In my opinion some points need to be improved.

1) Authors should use keywords that are not present in the title and have more impact.

2) in the introduction section describe more aspects of what is characterized as dementia in PD.

3) sections 2 and 3 describe that the decrease in BP at night is normal. I suggest including information as to why this decrease occurs while the person sleeps. What mechanisms are involved?

4) section 3 discusses the use of high-dose L-dopa. This drug is the most used in the treatment of PD symptoms and for this reason it has been used for many years as a chronic treatment. It is important to make clear to the reader what is considered a high dose. In addition, a parallel must be drawn between the side effects and the positive effects of using this drug.

5) Section 5: it seems that there is a relationship between diabetes/hypertension/cholesterolemia and PD and PD dementia. I suggest describing or elucidating some mechanism for this relationship to exist.

L209: 15 µmol/L. L of what? Blood?

Author Response

Dear Reviewer,

On behalf of my co-authors I would like to thank you for the valuable comments concerning the manuscript entitled: ‘Arterial blood pressure variability and other vascular factors contribution to the cognitive decline in Parkinson’s disease’. We did our best to revise the paper according to all comments and suggestions. Below is the point-by-point response to the comments with description of changes introduced into the text.

Reviewer 1.

Comments and Suggestions for Authors

Thank you for submitting the manuscript “Arterial Blood Pressure Variability and Other Vascular Factors Contribution to the Cognitive Decline in Parkinson’s Disease” to Molecules.

The review article is coherent and has a large volume of references, demonstrating that the authors have knowledge of the topic. In my opinion some points need to be improved.

1) Authors should use keywords that are not present in the title and have more impact.

The keywords have been changed according to the suggestion.

2) in the introduction section describe more aspects of what is characterized as dementia in PD.

We have added more information on the features of dementia in PD:

L35 Parkinson’s disease dementia mostly affects the executive and visuospatial functions, as well as attention. However, the impairment in memory and language functions are less pronounced than in Alzheimer’s disease (AD) [1].

3) sections 2 and 3 describe that the decrease in BP at night is normal. I suggest including information as to why this decrease occurs while the person sleeps. What mechanisms are involved?

We have added a paragraph shortly describing some of the mechanisms involved:

L79 This phenomenon is influenced by several factors, we briefly describe only some of them. First of all, photoreceptor cells in retina detect the environmental darkness, subsequently, the information is conveyed via the suprachiasmatic nucleus – the central oscillator – to the pineal gland, which reacts by producing melatonin. Melatonin de-creases BP by its impact on vasodilation, nitric oxide and norepinephrine levels. Accordingly, the pivotal role in the circadian BP pattern is exerted by the sympathetic nervous system with norepinephrine and epinephrine having the lowest levels in the evening and the highest in the morning. Sleep itself affects BP, decreasing its values especially during the deep sleep stages [12]. Similarly, physical activity may change the circadian rhythm – people who work at night, are likely to have blunted nocturnal drop in systolic BP. Dipping depends also on the BP liability upon dietary sodium intake and endothelial function [13].

4) section 3 discusses the use of high-dose L-dopa. This drug is the most used in the treatment of PD symptoms and for this reason it has been used for many years as a chronic treatment. It is important to make clear to the reader what is considered a high dose. In addition, a parallel must be drawn between the side effects and the positive effects of using this drug.

The information, on what was considered a high dose, has been added:

L152 According to a 5-year prospective study higher L-dopa dose was established as one of dementia risk factors in PD – the mean L-dopa dosage equivalent (LDE) was 514 mg per day, whereas non-demented patients received 316 mg of LDE per day [4].

 We have also briefly described the benefit-risk issue concerning dopaminergic medication in general:

L184 Nevertheless, despite many side effects, levodopa is still the most often prescribed antiparkinsonian medication due to its effectiveness against motor symptoms. Non-ergot derivatives are in turn beneficial in treating non-motor symptoms, e.g. depression, sleep disorders, and nocturnal akinesia [31]. In most cases positive effects of dopaminergic medication (L-dopa and dopamine agonists) surpass potential side effects in the cardiovascular system, however, the risk must be always individually estimated.

5) Section 5: it seems that there is a relationship between diabetes/hypertension/cholesterolemia and PD and PD dementia. I suggest describing or elucidating some mechanism for this relationship to exist.

An elucidation note has been added:

L229 The aetiology of cognitive impairment in PD presumably involves other than dopamine-dependent mechanisms, from which factors affecting the cerebrovascular status seem highly probable (Fig. 1) [61]. Hypertension, diabetes mellitus and hypercholesterolaemia lead to structural changes in the vessels [52], hypoperfusion, endothelial disorders, and altered blood-brain barrier permeability [70]. This results in cerebrovascular pathologies, seen as WMH – a marker of cognitive deterioration [52].

L209: 15 µmol/L. L of what? Blood?

We have added the necessary information: 15 µmol/L of blood (L265).

We hope that these changes to the manuscript will facilitate the decision to publish it.

Yours sincerely,

Marek DroĹşdzik

Reviewer 2 Report

This narrative review explains the role of arterial blood pressure variability and other vascular factors' contribution to the cognitive decline in PD. The article is interesting, and it addresses a current topic, however, there are a few areas that need to be added/improved before publishing.

  • Please add a table of genetic factors which are responsible for vascular and cognitive decline in PD.
  • Add an illustrative diagram that shows the role of a vascular factor in PD.
  • Add details of recent clinical trials on PD which targets vascular factors.
  • Please add a summarizing figure that supports the conclusions section.

Author Response

Dear Reviewer,

 On behalf of my co-authors I would like to thank you for the valuable comments concerning our manuscript entitled: ‘Arterial blood pressure variability and other vascular factors contribution to the cognitive decline in Parkinson’s disease’. We did our best to revise the paper according to the comments and suggestions. Below is the point-by-point response to the comments with description of changes introduced into the text.

Reviever 2.

This narrative review explains the role of arterial blood pressure variability and other vascular factors' contribution to the cognitive decline in PD. The article is interesting, and it addresses a current topic, however, there are a few areas that need to be added/improved before publishing.

  • Please add a table of genetic factors which are responsible for vascular and cognitive decline in PD.

We have added a table describing genetic factors (Table 2).

  • Add an illustrative diagram that shows the role of a vascular factor in PD.

We have added a diagram (Figure 1) explaining potential mechanisms.

  • Add details of recent clinical trials on PD which targets vascular factors.

We have included clinical trials that have targeted vascular factors in PD and have published the results – most of them covered the problem of OH and its treatment with droxidopa.

L70 Four clinical trials have addressed OH in conditions with dysautonomia, including PD, and have posted results by the time of this review (clinicaltrials.gov: NCT00738062, NCT00782340, NCT01176240, NCT00633880). In all of them droxidopa – a precursor of norepinephrine – was compared with placebo. The results showed some discrepancies, as only two of the trials established significant clinical changes favouring droxidopa in terms of increasing upright systolic BP, ameliorating symptoms associated with OH and improving daily life activities (i.e. standing, walking) (NCT00782340, NCT00633880) [10].

  • Please add a summarizing figure that supports the conclusions section.

We have added a figure (Figure 2).

We hope that these changes to the manuscript will facilitate the decision to publish it.

Yours sincerely,

Marek DroĹşdzik

Reviewer 3 Report

I have read with interest this paper summarizing some of the most important cardiovascular factors modulating the risk of cognitive impairment in Parkinson’s Disease. The authors especially focused on blood pressure variability, dopaminergic treatment, white matter hyperintensities, metabolic and vascular risk factors and, lastly genetic factors.

As a general comment I found the article well written, covering all the major literature evidences in both clinical and preclinical setting, discussing potential mechanisms of action and subsequent clinical implications in PD patients with suggestions for scientific knowledge gaps that need to be addressed. I have only some minor comments.

In the section discussing the role of dopaminergic treatment (section 3), the Authors really discuss only levodopa, and thus should be better name the subsection as “The role of levodopa treatment in neurocirculatory abnormalities and cognitive decline in Parkinson’s Disease”. Indeed, there is really paucity of literature on the subject, and I think that the inclusion of a statement underlining the needs of further studies on the association between other dopaminergic drugs and neurocirculatory abnormalities could act as a relevant suggestion for future readers.

In section 4 lines 171-172 I think that the sentence needs rephrasing, since it is not very clear. Moreover, I think that the authors should mention other two papers that faced the relationship between WMH and cognitive decline. In one paper, by Stojkovic et al. (Stojkovic T, Stefanova E, Soldatovic I, Markovic V, Stankovic I, Petrovic I, Agosta F, Galantucci S, Filippi M, Kostic V (2018) Exploring the relationship between motor impairment, vascular burden and cognition in Parkinson's disease. J Neurol 265:1320–1327.) the authors confirmed the association between WML and risk of PDD in a longitudinal cohort. In the other paper, by Nicoletti et al. (Nicoletti A, Luca A, Baschi R, Cicero CE, Mostile G, Davì M, La Bianca G, Restivo V, Zappia M, Monastero R. Vascular risk factors, white matter lesions and cognitive impairment in Parkinson's disease: the PACOS longitudinal study. J Neurol. 2021 Feb;268(2):549-558.) a significant association was found with WML and PDD, while it was only a trend with PD-MCI in a large longitudinal PD cohort.

In section 5, I think the authors could also include the aforementioned two papers when discussing the relationship with hypertension and cognitive impairment, since both of them offer another look on the association. Sotjkovic et al. confirming the role of vascular risk factors as predictors for both MCI and dementia in PD. Nicoletti et al, suggesting that hypertension acts as a risk factor for MCI only, contributing to an increased WML load that then modulates the risk of dementia.

Concerning the role of the Renin-angiotensin-aldosterone (RAA) system, I find the discussion really fascinating and a relevant starting point for future research. My suggestion is to include also a few lines on the potential relevance of drugs modulating the RAA system on the risk of future development of dementia, especially considering that Angiotensin-converting enzyme has been demonstrated to play a role in beta amyloid degradation, which constitutes part of the neuropathology of cognitive decline in PD (ref. an Dalen JW, Marcum ZA, Gray SL, Barthold D, Moll van Charante EP, van Gool WA, Crane PK, Larson EB, Richard E. Association of Angiotensin II-Stimulating Antihypertensive Use and Dementia Risk: Post Hoc Analysis of the PreDIVA Trial. Neurology. 2021 Jan 5;96(1):e67-e80.).

Author Response

Dear Reviewer,

On behalf of my co-authors I would like to thank you for the valuable comments concerning our manuscript entitled: ‘Arterial blood pressure variability and other vascular factors contribution to the cognitive decline in Parkinson’s disease’. We did our best to revise the paper according to the comments and suggestions. Below is the point-by-point response to the comments with description of changes introduced into the text.

Reviewer 3

I have read with interest this paper summarizing some of the most important cardiovascular factors modulating the risk of cognitive impairment in Parkinson’s Disease. The authors especially focused on blood pressure variability, dopaminergic treatment, white matter hyperintensities, metabolic and vascular risk factors and, lastly genetic factors.

As a general comment I found the article well written, covering all the major literature evidences in both clinical and preclinical setting, discussing potential mechanisms of action and subsequent clinical implications in PD patients with suggestions for scientific knowledge gaps that need to be addressed. I have only some minor comments.

In the section discussing the role of dopaminergic treatment (section 3), the Authors really discuss only levodopa, and thus should be better name the subsection as “The role of levodopa treatment in neurocirculatory abnormalities and cognitive decline in Parkinson’s Disease”. Indeed, there is really paucity of literature on the subject, and I think that the inclusion of a statement underlining the needs of further studies on the association between other dopaminergic drugs and neurocirculatory abnormalities could act as a relevant suggestion for future readers.

We have changed the name of the subsection and we have added a fragment on other dopaminergic drugs:

L166 There are some discrepancies on the role of dopamine agonists as well. Non-ergot derivatives of dopamine agonists possibly influence cardiovascular functions less, com-pared to ergot-derivatives [30]. Nevertheless, the side effects of non-ergot derivatives depend on the affinity to dopaminergic and α-adrenergic receptors [31]. Although non-ergot dopamine agonists were generally correlated with OH occurrence, rotigotine was showed to improve the abnormal 24-h BP pattern in PD patients [32]. On the other hand, rotigotine may increase the risk of the atrioventricular block. Pramipexole in turn, due to its high affinity towards α2-adrenergic receptors, may decrease the adrenergic tone and myocardial contractility, facilitating heart failure occurrence. Pramipexole may unmask a subclinical heart failure or exacerbate preexisting cardiovascular comorbidities [31]. Another non-ergot dopamine agonist – apomorphine – was suggested to pro-long QT interval; however, no causality has been established [31,33]. Undoubtedly, the aforementioned cardiovascular side effects influence BP and may lead to the impairments within the cerebrovascular functions, such as episodes of hypo- and hyperperfusion.

In section 4 lines 171-172 I think that the sentence needs rephrasing, since it is not very clear.

The sentence has been rewritten from:

According to the meta-analysis performed by Debette and Markus, WMH corre-lated with more dynamic decline in global cognitive performance, processing speed and executive function [35].

to:

L212 According to the meta-analysis performed by Debette and Markus, WMH corre-lated with more dynamic  predicted faster decline in global cognitive performance, processing speed and executive function [43].

Moreover, I think that the authors should mention other two papers that faced the relationship between WMH and cognitive decline. In one paper, by Stojkovic et al. (Stojkovic T, Stefanova E, Soldatovic I, Markovic V, Stankovic I, Petrovic I, Agosta F, Galantucci S, Filippi M, Kostic V (2018) Exploring the relationship between motor impairment, vascular burden and cognition in Parkinson's disease. J Neurol 265:1320–1327.) the authors confirmed the association between WML and risk of PDD in a longitudinal cohort. In the other paper, by Nicoletti et al. (Nicoletti A, Luca A, Baschi R, Cicero CE, Mostile G, Davì M, La Bianca G, Restivo V, Zappia M, Monastero R. Vascular risk factors, white matter lesions and cognitive impairment in Parkinson's disease: the PACOS longitudinal study. J Neurol. 2021 Feb;268(2):549-558.) a significant association was found with WML and PDD, while it was only a trend with PD-MCI in a large longitudinal PD cohort.

We have updated the Table 1 with the aforementioned papers.

In section 5, I think the authors could also include the aforementioned two papers when discussing the relationship with hypertension and cognitive impairment, since both of them offer another look on the association. Sotjkovic et al. confirming the role of vascular risk factors as predictors for both MCI and dementia in PD. Nicoletti et al, suggesting that hypertension acts as a risk factor for MCI only, contributing to an increased WML load that then modulates the risk of dementia.

We have included the outcome of the suggested papers in section 5:

L252 In a prospective research, hypertension was correlated with an increased risk of the MCI development in PD patients, whereas MCI and WMH predicted the conversion to PDD [52]. No impact of hypertension on the progression from MCI to PDD was established. Thus, it is likely that hypertension contributes to increasing WMH, which in turn modulates the risk of dementia. On the other hand, vascular factors (e.g. smoking status, body mass index, hypertension, and diabetes) – calculated together into Framingham General Cardiovascular Disease Risk Score – increased the risk of both MCI and dementia in PD [61].

Concerning the role of the Renin-angiotensin-aldosterone (RAA) system, I find the discussion really fascinating and a relevant starting point for future research. My suggestion is to include also a few lines on the potential relevance of drugs modulating the RAA system on the risk of future development of dementia, especially considering that Angiotensin-converting enzyme has been demonstrated to play a role in beta amyloid degradation, which constitutes part of the neuropathology of cognitive decline in PD (ref. an Dalen JW, Marcum ZA, Gray SL, Barthold D, Moll van Charante EP, van Gool WA, Crane PK, Larson EB, Richard E. Association of Angiotensin II-Stimulating Antihypertensive Use and Dementia Risk: Post Hoc Analysis of the PreDIVA Trial. Neurology. 2021 Jan 5;96(1):e67-e80.).

We have added a paragraph on this topic, with a reference to the suggested paper:

L337 A recent prospective study assessed the impact of several angiotensin II-stimulating (thiazides, dihydropyridine calcium channel blockers, AT1R blockers) and angiotensin II-inhibiting (ACE inhibitors, β-blockers, nondihydropyridine calcium channel blockers) antihypertensive drugs on the risk of dementia in older people (70-78 years at baseline) [87]. Subjects administrating AngII-stimulating antihypertensives had significantly lower risk of developing dementia compared to AngII-inhibiting drugs, independently of systolic BP or the baseline history of stroke, diabetes or CV diseases. As for the individual types of antihypertensives, the only significant decrease of dementia risk was associated with dihydropyridine calcium channel blockers. The beneficial role of upregulating AngII may result from the neuroprotection connected with AT2R. On the other hand, as ACE was showed to degrade amyloid-β (Aβ) in experimental models, the use of ACE inhibitors may promote Aβ plaque formation [87,88]. Regardless of the pivotal role of Aβ in AD, its deposition in PD was associated with cognitive decline [89]. However, in the aforementioned prospective study, the incidence rate of dementia in the group using ACE inhibitors did not differ from other types of antihypertensive drugs [87]. Moreover, Zhuang et al. showed in their meta-analysis that ACE inhibitors, as well as AT1R blockade, lowered the risk of AD, and AT1R blockade additionally decreased the risk of cognitive impairment associated with age [90]. Thus, the possible involvement of ACE in cognitive impairment and the effects exerted by ACE inhibitors seem to be far from being elucidated.

We hope that these changes to the manuscript will facilitate the decision to publish it.

Yours sincerely,

Marek DroĹşdzik

Round 2

Reviewer 1 Report

Thank you for submitting the requested manuscript corrections to Molecules. The quality of the manuscript has improved a lot and in my opinion it can be accepted for publication in this form.